# REPRESENTATION LEARNING FOR IMPROVED INTERPRETABILITY AND CLASSIFICATION ACCURACY OF CLINICAL FACTORS FROM EEG

**Garrett Honke**[*][†]
X, the Moonshot Factory
Mountain View, CA

**Irina Higgins**[*][†]
DeepMind
London, UK

**Nina Thigpen**[†]
X, the Moonshot Factory
Mountain View, CA, USA

**Vladimir Miskovic**[†]
X, the Moonshot Factory
Mountain View, CA, USA

**Katie Link**[†]
X, the Moonshot Factory
Mountain View, CA, USA

**Sunny Duan**[†]
DeepMind
Mountain View, CA, USA

**Pramod Gupta**[†]
X, the Moonshot Factory
Mountain View, CA, USA

**Julia Klawohn**[‡]
Florida State University
Tallahassee, FL, USA

**Greg Hajcak**[§]
Florida State University
Tallahassee, FL, USA

## ABSTRACT

Despite extensive standardization, diagnostic interviews for mental health disorders encompass substantial subjective judgment. Previous studies have demonstrated that EEG-based neural measures can function as reliable objective correlates of depression, or even predictors of depression and its course. However, their clinical utility has not been fully realized because of 1) the lack of automated ways to deal with the inherent noise associated with EEG data at scale, and 2) the lack of knowledge of which aspects of the EEG signal may be markers of a clinical disorder. Here we adapt an unsupervised pipeline from the recent deep representation learning literature to address these problems by 1) learning a disentangled representation using $\beta$-VAE to denoise the signal, and 2) extracting interpretable features associated with a sparse set of clinical labels using a Symbol–Concept Association Network (SCAN). We demonstrate that our method is able to outperform the canonical baseline classification method on a number of factors, including participant age and depression diagnosis. Furthermore, our method recovers a representation that can be used to automatically extract denoised Event Related Potentials (ERPs) from novel, single EEG trajectories, and supports fast supervised re-mapping to various clinical labels, allowing clinicians to re-use a single EEG representation regardless of updates to the standardized diagnostic system. Finally, single factors of the learned disentangled representations often correspond to meaningful markers of clinical factors, as automatically detected by SCAN, allowing for human interpretability and post-hoc expert analysis of the recommendations made by the model.

## 1 INTRODUCTION

Mental health disorders make up one of the main causes of the overall disease burden worldwide (Vos et al., 2013), with depression (e.g., Major Depressive Disorder, MDD) believed to be the second leading cause of disability (Lozano et al., 2013; Whiteford et al., 2013), and around 17% of the population experiencing its symptoms at some point throughout their lifetime (McManus et al., 2016; 2009; Kessler et al., 1993; Lim et al., 2018). At the same time diagnosing mental health disorders has many well-identified limitations (Insel et al., 2010). Despite the existence of diagnostic manuals

---

[*]Equal contribution

[†]{ghonk,irinah,nthigpen,vmiskovic,katielink,sunnyd,pramodg}@google.com

[‡]JK is now at Humboldt-Universität zu Berlin, Berlin, Germany; julia.klawohn@hu-berlin.de

[§]greg.hajcak@med.fsu.edu

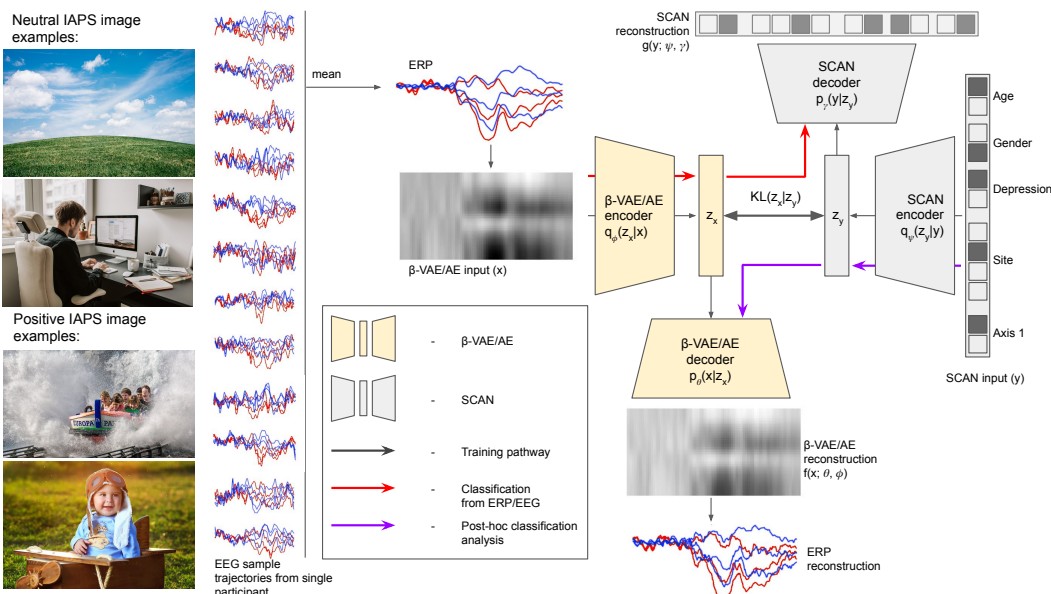

Figure 1: Pipeline schematic. Participants are presented with images from the International Affective Picture System (IAPS). EEG trajectories recorded from the same participant over multiple trials are averaged to create ERPs. Each EEG sample consists of 256 time samples (-248-772 ms, where 0 is stimulus onset time) of stimulus-locked activity recorded from three sites as participants view either neutral (red) or positive (blue) IAPS images. ERP responses to positive and negative images are concatenated and normalised between [0, 1] across all channels simultaneously. The resulting "images" are used to train $\beta$-VAE or AE. A well disentangled pre-trained $\beta$-VAE model is used to train SCAN. Each SCAN training example consists of a 5-hot binary classification label $\mathbf{y}$ presented to the SCAN encoder, and its corresponding EEG "image" $\mathbf{x}$ presented to the $\beta$-VAE encoder. $\beta$-VAE weights are fixed during SCAN training. To obtain classification, an EEG "image" is presented to the $\beta$-VAE encoder, then the inferred $\mathbf{z}_x$ means are fed through the SCAN decoder, where a per-class softmax is applied to obtain the predicted label (red pathway). To analyse SCAN classification decisions, a 1-hot binary vector is fed into SCAN encoder. Samples from the inferred distribution $\mathbf{z}_y$ are then fed through the $\beta$-VAE decoder to visualise the corresponding ERP reconstructions (purple pathway).

like Structured Clinical Interview for Diagnostic and Statistical Manual of Mental Disorders (SCID) (DSM-V, 2013), diagnostic consistency between expert psychiatrists and psychologists with decades of professional training can be low, resulting in different diagnoses in upwards of 30% of the cases (Cohen's Kappa = 0.66) (Lobbestael et al., 2011). Even if higher inter-rater reliability was achieved, many psychological disorders do not have a fixed symptom profile, with depression alone having many hundreds of possible symptom combinations (Fried & Nesse, 2015). This means that any two people with the same SCID diagnosis can exhibit entirely different symptom expressions. This is a core challenge for developing an objective, symptom-driven diagnostic tool in this domain.

Electroencephalography (EEG) is a measurement of post-synaptic electrical potentials that can be taken non-invasively at the scalp. EEG signals can function as important biomarkers of clinical disorders (Hajcak et al., 2019) but they are difficult to clean and interpret at scale. For example, components of the EEG signal can often significantly overlap or interfere with each other. Furthermore, nearby electronics, line noise, hardware quality, signal drift and other variations in the electrode–scalp connection can all distort the recorded EEG signal. Hence, the extraction of EEG data of sufficient quality is usually a laborious, semi-automated process executed by lab technicians with extensive training. A typical EEG analysis pipeline consists of collecting EEG recordings evoked from a large number of stimulus presentations (trials) in order to have sufficient data to average out the noise. Independent Components Analysis (ICA) is often used to visually identify and remove the component that corresponds to eye blinks (Delorme & Makeig, 2004; Makeig et al., 2004; Jung et al., 2000) (although see Weber et al. (2020); Nolan et al. (2010) as examples of fully automated artifact removal pipelines) . This can be followed by a trial rejecton stage where anomalous trials are identified and removed from the EEG data scroll, sometimes also through visual examination. The cleaned up EEG recordings from a large number of trials are then averaged to produce an Event Related Potential (ERP) (Luck, 2012). This allows a clinician to extract specific ERP components relevant to the clinical factor of interest, average out the event-locked activity within them, and then either perform a statistical group comparison, or—in

the case of the diagnosis classification goal—apply an off-the-shelf classifier, like Logistic Regression (LR) to obtain the final diagnostic results. Some more advanced classification approaches might include Support Vector Machines (SVM), Linear Discriminant Analysis (LDA), or Random Forest (RF) (Parvar et al., 2015; Güler & Übeyli, 2007; Subasi & Gursoy, 2010; Tomioka et al., 2007; Bashivan et al., 2016).

To summarise, EEG recordings are noisy measures of electric activity from across the brain. There is evidence that these signals are useful as markers of depression, but we lack understanding of what aspects of depression they index. Furthermore, the field of clinical psychopathology still lacks consensus on the etiopathogenesis of mental health disorders, which means that there is no such thing as the "ground truth" diagnostic labels. Hence, while EEG is routinely used to diagnose conditions like epilepsy (Smith, 2005), memory (Stam et al., 1994) or sleep disorders (Karakis et al., 2012), its promise for being a reliable diagnostic tool for clinical conditions like depression has not been fully realised so far. In order to make EEG a viable diagnostic tool for a broader set of clinical conditions it is important to have an automated pipeline for extracting the relevant interpretable biomarker correlates from the (preferably individual trial) EEG data in a robust manner. Furthermore, this process should not depend fully on diagnostic labels which are often subjective and at best represent highly heterogeneous classes.

Recent advances in deep learning have prompted research into end-to-end classification of EEG signal using convolutional and/or recurrent neural networks (Bashivan et al., 2016; Mirowski et al., 2009; Cecotti & Graser, 2011; Güler et al., 2005; Wang et al., 2018; Farahat et al., 2019; Solon et al., 2019; Cecotti et al., 2014), holding the promise of automated extraction of relevant biomarker correlates. However, deep classifiers operate best in the big data regime with clean, well-balanced ground truth classification targets. In contrast, even the largest of EEG datasets typically contain only a few hundred datapoints, and the classification labels are subjective, noisy and unbalanced, with the majority of the data coming from healthy control participants. Hence, in order to utilise the benefits of deep learning but avoid the pitfalls of over-reliance on the classification labels, we propose a two-step pipeline consisting of unsupervised representation learning, followed by supervised mapping of the pre-trained representation to the latest version of the available diagnostic labels. The hope is that the unsupervised learning step would denoise the input signal and extract the broad statistical regularities hidden in it thus serving as an alternative for the existing automatic EEG pre-processing pipelines (Weber et al., 2020; Nolan et al., 2010) while minimising the need for a priori knowledge , resulting in a representation that can continue to be useful even if the label taxonomy evolves.

Recently great progress has been made in the field of deep unsupervised representation learning (Roy et al., 2019; Devlin et al., 2018; Brown et al., 2020; Chen et al., 2020b; Grill et al., 2020; Chen et al., 2020a; Higgins et al., 2017; Burgess et al., 2019). Disentangled representation learning is a branch of deep unsupervised learning that produces interpretable factorised low-dimensional representations of the training data (Bengio et al., 2013; Higgins et al., 2017). Given the requirement for model interpretability in our use-case, we use Beta Variational Autoencoders ($\beta$-VAE) (Higgins et al., 2017)—one of the state of the art unsupervised disentangled representation learning methods—to discover low-dimensional disentangled representations of the EEG data. We then train the recently proposed Symbol–Concept Association Network (SCAN) (Higgins et al., 2018) to map the available classification labels to the representations learnt by $\beta$-VAE (see Fig. 1). We demonstrate that our proposed pipeline results in better classification accuracy than the typical approach for extracting a known ERP pattern for use as a biomarker—a process that is often heavily influenced by a priori knowledge. This holds true when predicting a number of factors, including age, gender, and depression diagnosis. Furthermore, SCAN is able to produce arguably interpretable classification recommendations, whereby its decisions on different clinical factors are grounded in a small number (often single) latent dimensions of the $\beta$-VAE, allowing the clinicians an opportunity to interpret the recommendations produced by SCAN, and visualise what aspects of the EEG signal are associated with the classification decision post-hoc. This opens up the opportunity to use our proposed pipeline as a tool for discovering new EEG biomarkers. We validate this by "re-discovering" a known biomarker for depression. Finally, we demonstrate that once a $\beta$-VAE is pre-trained on ERP signals, it can often produce ERP-like reconstructions even when presented with single noisy EEG trajectories. Furthermore, the representations inferred from single EEG trials produce good classification results, still outperforming the canonical baseline method. This suggests that once a good disentangled representation is learnt, the model can be used online as new EEG data is being recorded, thus lowering the burden of keeping potentially vulnerable participants in the lab for extended recording sessions.

## 2 METHODS

### 2.1 DATA

**Anhedonia.** This work targets one of the two cardinal symptoms of depression—anhedonia. Anhedonia is the lack of pleasure and/or interest in previously pleasurable stimuli and activities (DSM-V, 2013). One established approach for objectively quantifying this symptom is the use of EEG to measure neural responses elicited by emotionally salient visual stimuli. Research in this domain has uncovered a stereotyped neural activation pattern in healthy control participants, where emotionally-salient stimuli evoke a Late Positive Potential (LPP) in ERPs—the averaged timeseries of stimulus time-locked EEG recordings. This pattern has been identified as a potential biomarker for depression because (on average) this positive deflection in amplitude is attenuated or absent in individuals who exhibit symptoms of anhedonia (Brush et al., 2018; Foti et al., 2010; Klawohn et al., 2020; MacNamara et al., 2016; Weinberg et al., 2016; Weinberg & Shankman, 2017).

**Participants.** The data were collected as part of a mental health study across multiple laboratory sites. The multi-site aspect of the study meant that more data could be pooled together, however, it also meant that the data was noisier. Participants ($N = 758$, age$_{\bar{X}} = 16.7$, age$_{\text{range}} = [11.0, 59.8]$, 398 female) were samples of healthy controls ($n_{\text{HC}} = 485$) and people diagnosed with depression (among other mental illnesses) (see Sec. A.1.1 and Tbl. A1 for further breakdown).

**Stimuli and Experimental Design.** Participants were shown a series of 80 images from the International Affective Picture System (IAPS) (Lang et al., 2008) presented in random order up to 40 times each. The images varied in valence: either positive (affiliative scenes or cute animals designed to elicit the LPP ERP component), or neutral (objects or scenes with people). Each image trial consisted of a white fixation cross presented for a random duration between 1000-2000 ms (square window) followed by a black and white image presented for 1000 ms.

**EEG Preprocessing.** While participants completed the picture viewing task, EEG was continuously recorded. Each picture trial was then segmented to contain a 200 ms pre-stimulus baseline and a 1200 ms post-stimulus interval. The raw EEG signal was digitized, bandpass filtered and cleared of the eye movement artifacts and anomalous trials as described in Sec. A.1.2.

**Classification labels.** The following classification labels were used in this study: age (adult or child), gender (male or female), study site, and the presence or absence of two clinical conditions: depression diagnosis and a broader Axis 1 disorder diagnosis. All classification labels were binary, apart from study site, which contained four possible values corresponding to four different sites where the data were collected. Participants 18 years of age and older were classified as adults. Gender was classified based on self-reported values. Positive depression labels ($n = 110$) include all participants that were diagnosed with Major Depressive Disorder (MDD), Persistent Depressive Disorder (PDD), and depressive disorder NOS (not-otherwise-specified) by expert clinicians through a clinical interview (e.g., SCID for adults, KSADS for children). Axis 1 is a broad category consisting of the most prevalent psychological disorders in the population (now discontinued in DSM-V) that excludes intellectual disabilities and personality disorder (DSM-V, 2013). Positive Axis 1 labels ($n = 273$) encompassed all participants with positive depression labels plus individuals diagnosed with Cyclothemia, Dysthemia, anxiety disorders, mood disorders (e.g., Bipolar Disorder), panic disorders, and substance and eating disorders (all of which are sparse). The Axis 1 class is provided to compare model behavior on a transdiagnostic measurement of psychopathology.[1] While recruitment for the study was primarily focused on depression, the SCID produces a large set of diagnostic decisions, and we collapsed this sparser set of positive diagnoses into the existing Axis 1 DSM-IV superordinate category. We include this label in modeling and analysis to maximize the number of positive labels for training and evaluation and to give the reader a sense of what the algorithm may have learned that is generalizable across disorders—akin to the $P$ factor (Caspi et al., 2014).

---

[1]Disorders in this class that are present in the data include Major Depressive Disorder, Persistent Depressive Disorder, Depression NOS, Cyclothemia, Dysthemia, Bipolar I, Bipolar II, Bipolar NOS, Mania, Hypomania, Agoraphobia, Social Phobia, Separation Anxiety, Generalized Anxiety Disorder, Panic Disorder, Panic Disorder with Agoraphobia, Anorexia, Bulimia, Eating disorder NOS, Alcohol Abuse, Alcohol Dependence, and Substance Abuse and Substance Dependence disorders.

## 2.2 REPRESENTATION LEARNING

**Canonical LPP analysis baseline** The canonical approach for extracting the Late Positive Potential (LPP) effect serves as the baseline for this work. The LPP effect is calculated as the average amplitude difference between ERP waveforms (i.e., averaged stimulus time-locked EEG segments) evoked from emotionally-salient and neutral stimuli. Before the delta was calculated, each ERP was normalised with respect to the baseline average activity within the 100 ms window preceding stimulus onset. Finally, the normalised delta signal within the 300–700 ms window after stimulus onset was averaged to provide the canonical baseline LPP representation.

**Autoencoder** An AutoEncoder (AE) is a deep neural network approach for non-linear dimensionality reduction (Hinton & Salakhutdinov, 2006; Baldi, 2012). A typical architecture consists of an encoder network, which projects the input high-dimensional data $\mathbf{x}$ into a low-dimensional representation $\mathbf{z}$, and a decoder network that is the inverse of the encoder, projecting the representation $\mathbf{z}$ back into the reconstruction of the original high-dimensional data $f(\mathbf{x}; \phi,\theta)$, where $\phi$ and $\theta$ are the parameters of the encoder and decoder respectively (see Fig. 1 for model schematic). AE is trained through backpropagation (Rumelhart et al., 1986) using the reconstruction objective:

$$\mathcal{L}_{AE} = \mathbb{E}_{p(\mathbf{x})} \, ||f(\mathbf{x}; \phi,\theta) - \mathbf{x}||^2$$

The input to the AE in our case is a 256x6 "image" of the EEG signal (see Fig. 1), where 256 corresponds to the 1024 ms of the recorded EEG trajectory sampled at 250 Hz and pre-processed as described in Sec. A.1.2, and 6 corresponds to three electrodes per each of the two image valence conditions (i.e., stimulus classes): neutral and positive. These input images were further normalised to the [0, 1] range across all channels before being presented to the AE.

The AE was parametrised to have two convolutional encoding layers with 32 filters each of size 6, with strides of step size 2 along the time axis, followed by a single fully connected layer of size 128, projecting into a 10-dimensional representation $\mathbf{z}$. The decoder was the inverse of the encoder. Overall the model consisted of around 106,752 parameters. The model had ReLU activations throughout, and was optimised using Adam optimizer with learning rate of 1e-4 over 1 mln iterations of batch size 16.

**$\beta$-Variational Autoencoder** A $\beta$-Variational Autoencoder ($\beta$-VAE) (Higgins et al., 2017) is a generative model that aims to learn a disentangled latent representation $\mathbf{z}$ of input data $\mathbf{x}$ by augmenting the Variational Autoencoder (VAE) framework (Rezende et al., 2014; Kingma & Welling, 2014) (see Sec.A.2.1 for more details) with an additional $\beta$ hyperparameter. Intuitively, a disentangled representation is a factorised latent distribution where each factor corresponds to an interpretable transformation of the training data (e.g. in a disentangled representation of a visual scene, individual factors may represent a change in lighting or object position). A neural network implementation of a $\beta$-VAE consists of an inference network (equivalent to the AE encoder), that takes inputs $\mathbf{x}$ and parameterises the (disentangled) posterior distribution $q(\mathbf{z}|\mathbf{x})$, and a generative network (equivalent to the AE decoder) that takes a sample from the inferred posterior distribution $\hat{\mathbf{z}} \sim \mathcal{N}(\mu(\mathbf{z}|\mathbf{x}),\sigma(\mathbf{z}|\mathbf{x}))$ and attempts to reconstruct the original image (see Fig. 1). The model is trained through a two-part loss objective:

$$\mathcal{L}_{\beta-VAE} = \mathbb{E}_{p(\mathbf{x})} \big[ \, \mathbb{E}_{q_\phi(\mathbf{z}|\mathbf{x})}[\log p_\theta(\mathbf{x}|\mathbf{z})] - \beta KL(q_\phi(\mathbf{z}|\mathbf{x}) \, || \, p(\mathbf{z})) \, \big]$$

where $p(\mathbf{x})$ is the probability of the input data, $q(\mathbf{z}|\mathbf{x})$ is the learnt posterior over the latent units given the data, $p(\mathbf{z})$ is the unit Gaussian prior with a diagonal covariance matrix $\mathcal{N}(\mathbf{0},\mathbb{I})$, $\phi$ and $\theta$ are the parameters of the inference (encoder) and generative (decoder) networks respectively, and $\beta$ is a hyperparameter that controls the degree of disentangling achieved by the model during training. Intuitively, the objective consists of the reconstruction term (which aims to increase the log-likelihood of the observations) and a compression term (which aims to reduce the Kullback-Leibler (KL) divergence between the inferred posterior and the prior). Typically a $\beta > 1$ is necessary to achieve good disentangling, however the exact value differs for different datasets. In order to find a good value of $\beta$ to disentangle our EEG dataset, we perform a hyperparameter search, training ten models with different random initialisations for each of the ten values of $\beta \in [0.075, 2.0]$ sampled uniformly. Well disentangled $\beta$-VAE models were selected using the Unsupervised Disentanglement Ranking (UDR) score (Duan et al., 2019) described in Sec. A.4 (see Fig. A1 for a visualisation of the resulting UDR

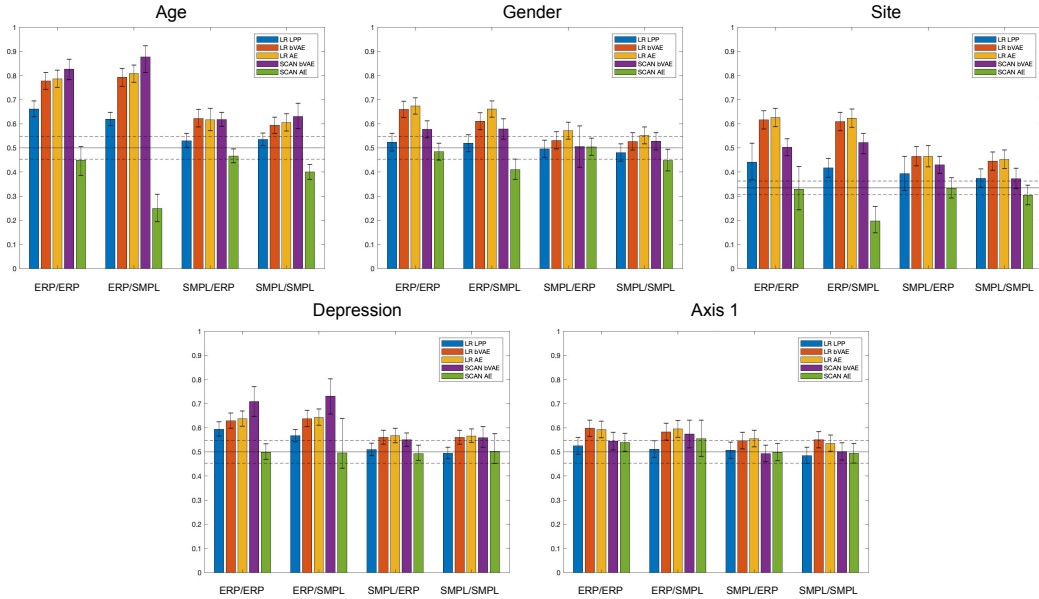

Figure 2: Posterior distribution of balanced classification accuracy (mean, error bars - 95% confidence interval) (Carrillo et al., 2014; Brodersen et al., 2010). ERP - averaged trajectories, SMPL - single sampled EEG trials. LPP - canonical late positive potential baseline representation. LR - L2 regularised logistic regression (see Tbl. A2 for other baseline classifiers). Models trained on single EEG trajectories (SMPL) are tested on different single EEG trajectories in the SMPL/SMPL train/test case. Black horizontal line - chance accuracy, horizontal dashed lines - 95% confidence interval calculated according to Müller-Putz et al. (2008).

scores). All $\beta$-VAE models had the same architecture as the AE[2], and were trained in the same manner and on the same data that was pre-processed in the same way to lie in the [0, 1] range .

## 2.3 CLASSIFICATION

**Baseline classifiers** To evaluate the quality of a representation in terms of how useful it is for classifying different clinical factors, we applied a range of baseline classification algorithms: Support Vector Machine (SVM), Random Forest (RF), Logistic Regression (LR) and Linear Discriminant Analysis (LDA) (see Sec. A.3 for details). For all classification results we report the posterior distribution of balanced accuracy (Carrillo et al., 2014; Brodersen et al., 2010). Balanced accuracy was chosen, because it correctly matches chance accuracy even for unbalanced datasets. Chance accuracy and its confidence bounds were calculated according to Müller-Putz et al. (2008).

**Symbol–Concept Association Network** The baseline classifiers described above produced uninterpretable decisions. This is undesirable if we were to have a chance at discovering new clinical biomarkers in the EEG data. To address this interpretability challenge, we leverage a recent model proposed for visual concept learning in the machine learning literature—the Symbol–Concept Association Network (SCAN) (Higgins et al., 2018). While SCAN was not originally developed with the classification goal in mind, it has desirable properties to utilise for the current application. In particular, it is able to automatically discover sparse associative relationships between discrete symbols (5-hot classification labels in our case, see Fig. 1) and continuous probability distributions of the disentangled posterior representation discovered by a trained $\beta$-VAE model. Furthermore, the associative nature of the grounding used to train SCAN allows it to deal with noisy data gracefully, and to learn successfully from a small number of positive examples and from highly unbalanced datasets. SCAN is in effect another VAE model. In our case it takes 5-hot classification labels $\mathbf{y}$ as input, and aims to reconstruct them from the inferred posterior $q(\mathbf{z}|\mathbf{y})$ (see Fig. 1 for more details). To train SCAN the original VAE objective is augmented with an additional KL term that aims to ground the SCAN posterior in the posterior of a pre-trained $\beta$-VAE model:

$$\mathcal{L}_{SCAN} = \mathbb{E}_{p(\mathbf{y})} \left[ \mathbb{E}_{q_\psi(\mathbf{z}_y|\mathbf{y})} [\log p_\gamma(\mathbf{y}|\mathbf{z}_y)] - KL(q_\psi(\mathbf{z}_y|\mathbf{y}) \,||\, p(\mathbf{z}_y)) \right] - KL(q_\phi(\mathbf{z}_x|\mathbf{x}) \,||\, q_\psi(\mathbf{z}_y|\mathbf{y}))]$$

[2]Note that the AE can be seen as a special case of the $\beta$-VAE with $\beta = 0$.

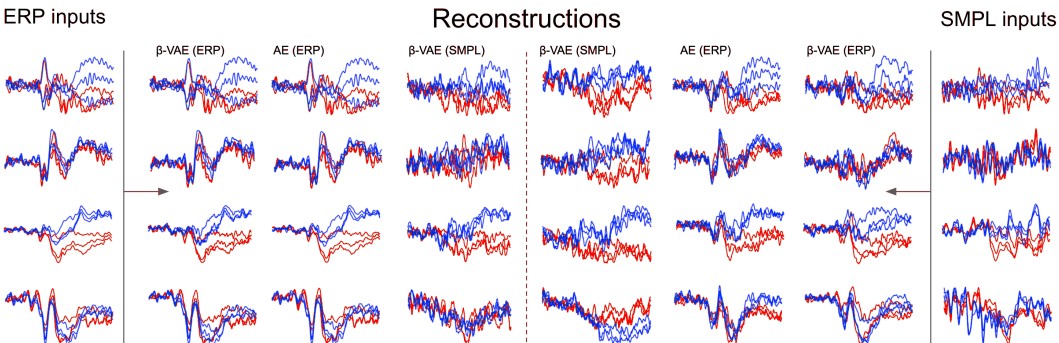

Figure 3: Reconstructions (middle six columns) by two disentangled $\beta$-VAEs pre-trained on either ERPs or single EEG sample trajectories (SMPL), as well as an AE pre-trained on ERPs. Each model reconstructs either single EEG trajectories (SMPL inputs) or the corresponding ERPs (ERP inputs). Models pre-trained on ERPs are able to reconstruct ERP-like trajectories even from single EEG samples, as demonstrated by the closer similarity of the $\beta$-VAEs (ERP) and AE (ERP) reconstructions to ERP inputs compared to SMPL inputs, regardless of whether ERPs or single EEG samples are reconstructed.

where $\psi$ and $\gamma$ are the parameters of the SCAN encoder and decoder respectively, $q_\psi(\mathbf{z}_y|\mathbf{y})$ is the posterior of SCAN inferred from a 5-hot label $\mathbf{y}$, and $q_\phi(\mathbf{z}_x|\mathbf{x})$ is the posterior of the pre-trained $\beta$-VAE model inferred from an EEG "image" corresponding to the label presented to SCAN. Note that the $\beta$-VAE weights are not updated during SCAN training. The extra grounding term $KL(q_\phi(\mathbf{z}_x|\mathbf{x}) \,||\, q_\psi(\mathbf{z}_y|\mathbf{y}))$ allows SCAN to discover an associative relationship between a subset of disentangled factors and each 1-hot dimension of the given label.

We paramterised SCAN encoder and decoder as MLPs with two hidden layers of size 128, ReLU non-linearities and a 10-dimensional posterior to match the dimensionality of the $\beta$-VAE representation. The model resulted in around 22,016 parameters. Like the other models, SCAN was trained with batch size 16, over 1 mln iterations and with Adam optimizer with learning rate of 1e-4.

## 3  RESULTS

**Deep representation learning improves classification accuracy.**  We evaluated the classification accuracy for predicting participant age, gender, study site, and the presence or absence of two clinical labels: depression and Axis 1. We first obtained the canonical LPP representations, and applied Support Vector Machine (SVM), Random Forest (RF), Logistic Regression (LR) and Linear Discriminant Analysis (LDA) to obtain balanced classification accuracy. Table A2 and Figure A2 show that LR produced some of the best results overall for the baseline, hence we report only LR results for the $\beta$-VAE and AE representations in the main text (see Tables A3-A4 and Figures A3-A4 for the other classifiers applied to the $\beta$-VAE and AE representations). Figures 2 and 5 (ERP/ERP train/test condition) demonstrate that representations extracted through $\beta$-VAE and AE pretraining resulted in higher overall classification accuracy than those obtained through the baseline canonical LPP pipeline (see also Table A5 ERP/ERP cells). This effect holds across all classification tasks, including depression and Axis 1 diagnoses. Furthermore, the pattern of classification results is in line with what might have been expected by the expert clinicians, whereby age and depression classification accuracy is significantly higher than chance, while gender is harder to decode from the EEG signal.

A similar pattern appears to hold in most cases when all the models are trained on single EEG trials instead of ERPs (SMPL/ERP train/test in Figures 2 and 5, see also Table A5). While the maximum classification accuracy drops for all models when trained from the noisier single EEG trials rather than ERPs, the classification accuracy obtained from $\beta$-VAE and AE representations is still often higher than that obtained from the LPP baseline.

**Classification based on deep representations generalises better to single EEG trials.**  One possible clinical application of the proposed classification pipeline is to enable online diagnosis recommendations from single EEG trials. Hence, we tested how well the classification accuracy of the different representations generalises to novel single EEG trial trajectories (*/SMPL in Figures 2 and 5, see also SMPL columns in Tbl. A5). As expected, the performance often drops when classifying the noisy single EEG trajectories, regardless whether the representations were pre-trained using

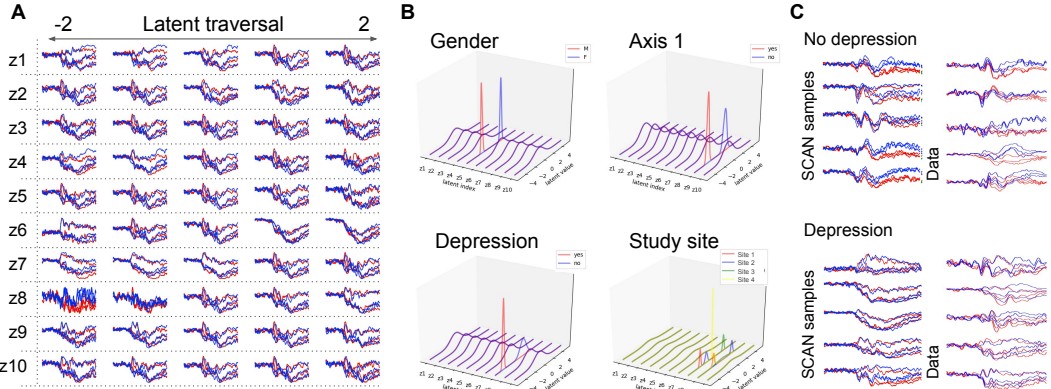

Figure 4: **A**: Latent traversals of a pre-trained disentangled $\beta$-VAE. Each row reconstructs an ERP trajectory as the value of each latent dimension is traversed between [-2, 2] while keeping the values of all other latents fixed. **B**: Inferred distributions for each of the 10 latent dimensions of a SCAN trained with the $\beta$-VAE shown in A. Each subplot shows the result of SCAN inference on a set of 1-hot symbols corresponding to all the different conditions of a single factor (e.g. M/F for Gender). Different factors have automatically become associated with different (often single) latent dimensions. For example, the presence (red) or absence (blue) of Axis 1 diagnosis is represented by the narrow inferred Gaussian distributions centered around different opposite values of latent $z_9$, while other latent dimensions are defaulted back to the unit Gaussian prior. **C**: five samples from each of the two conditions associated with the depression factor from the SCAN shown in B, as well as corresponding ERP samples from the training dataset . Red - neutral responses, blue - pleasant responses from each of the three EEG sensors. SCAN has automatically discovered that the presence of depression is associated with the lack of clear separation between responses in the two valence conditions—anhedonia. Its samples are qualitatively similar to the samples from the dataset.

ERPs (ERP/SMPL) or single EEG samples (SMPL/SMPL, note that we tested the models with different single EEG samples to those used for training). However, the classification accuracy is still often significantly higher for deep representations compared to the LPP baseline. This suggests that replacing the more manual canonical LPP pipeline with deep representation learning can allow for both better training data efficiency and a reduction in time that the (potentially vulnerable) participants have to spend in the lab by up to 37x, which is the average number of trials per condition that made it through EEG pre-processing in our dataset and were used for generating ERPs.

**Deep representation learning reconstructs ERPs from single EEG trials.** Since $\beta$-VAE and AE had good classification accuracy when presented with single EEG samples, we tested whether they could also reconstruct ERPs from single EEG trials. Figure 3 shows that this is indeed the case— reconstructions produced by the pre-trained models from single noisy EEG samples look remarkably similar to those produced from the corresponding ERPs (also see Figs. A5-A7 for more examples). Note that this only holds true for models trained using ERP data and not those trained on single EEG samples.

**Disentangled representations allow for interpretable classification.** To obtain SCAN classification results, we inferred the posterior $q_\phi(\mathbf{z}_x|\mathbf{x})$ of a pre-trained $\beta$-VAE in response to an EEG "image" $\mathbf{x}$. We then used the pre-trained SCAN decoder to obtain the reconstructed label logits $p_\gamma(\mathbf{y}|\mu(\mathbf{z}_x))$ using the $\beta$-VAE posterior mean as input (Fig. 1, red path). Finally, we applied softmax over the produced logits to obtain the predicted 5-hot label for the EEG "image". When SCAN was trained on top of a well disentangled $\beta$-VAE, it was often able to outperform the canonical LPP baseline in terms of classification accuracy (see Figure 2 and Table A5, SCAN+$\beta$-VAE). This is despite the fact that SCAN was not developed with the classification goal in mind. Note, however, that SCAN can only work when trained with disentangled representations. Indeed, SCAN classification performance was in most conditions not distinguishable from chance when trained on top of entangled AE representations (see Figure 2 and Table A5, SCAN+AE). To further confirm the role of disentanglement, we calculated Spearman correlation between the quality of disentanglement as measured by the UDR score and the final SCAN classification accuracy. Table 1 shows significant correlation for age, depression and Axis 1 diagnoses, suggesting that on average these factors were classified better if representations were more disentangled. The same, however, is not the case for gender or study site. This implies that some of the better disentangled models did not contain information necessary for classifying these factors. Such information loss is

| Age | | Gender | | Site | | Depression | | Axis 1 | |
|---|---|---|---|---|---|---|---|---|---|
| $r$ | $p$ | $r$ | $p$ | $r$ | $p$ | $r$ | $p$ | $r$ | $p$ |
| 0.28 | .01 | -0.1 | .31 | 0.19 | .06 | 0.33 | <.001 | 0.31 | <.001 |

Table 1: Spearman correlation between disentanglement quality (as approximated by UDR score) of trained $\beta$-VAE models ($n$=100) and balanced classification accuracy from their corresponding SCAN models.

in fact a known trade-off of $\beta$-VAE training, with more disentangled $\beta$-VAE models often compromising on the informativeness of the learnt representation due to the increased compression pressure induced by the higher $\beta$ values necessary to achieve disentanglement (Higgins et al., 2017; Duan et al., 2019).

When trained on top of well disentangled $\beta$-VAE models, SCAN decisions were not only accurate, but they were also based on a small number of disentangled dimensions. This is unlike the LR classifier, which obtained average sparsity of just 3.25% compared to the 87.5% for SCAN, even when L1 regularisation was used. Furthermore, the disentangled dimensions used by SCAN were often arguably interpretable, hence making SCAN classification decisions amenable to post-hoc analysis. Figure 4B visualises the inferred SCAN posterior when presented with 1-hot labels corresponding to the male or female gender, one of the four study sites, and the presence or absence of depression and Axis 1 diagnoses. In most cases, SCAN was able to associate the label with a single latent dimension in the pre-trained $\beta$-VAE (e.g., gender is represented by latent dimension $z_5$), and the different values of the same class label corresponded to disjoint narrow Gaussians on those latents (e.g., male gender is represented with a Gaussian $\mathcal{N}(0.89,\ 0.49)$, while female gender is represented by a Gaussian $\mathcal{N}(-0.82,\ 0.52)$, both on $z_5$). We can also visualise what the different $\beta$-VAE latents have learnt to represent by plotting their traversals as shown in Figure 4A (see more in Figs. A8-A10).

Finally, we can also sample from the SCAN posterior $\hat{\mathbf{z}}_y \sim \mathcal{N}(\mu(\mathbf{z}_y|\mathbf{y}), \sigma(\mathbf{z}_y|\mathbf{y}))$ and reconstruct the resulting ERPs using the $\beta$-VAE decoder $p_\theta(\mathbf{x}|\hat{\mathbf{z}}_y)$ (Fig. 1, purple path). Such samples for the depression label are visualised in Figure 4C. It is clear that the reconstructed ERP samples corresponding to participants with a positive depression diagnosis contain no difference between the positive and neutral trials (overlapping blue and red lines), while those corresponding to participants without a positive depression diagnosis do have an obvious gap between the responses to positive and neutral trials. Hence, we were able to "re-discover" an objective biomarker for the symptom of anhedonia (and depression generally), thus opening up the potential for discovering new biomarkers hidden in EEG data in the future.

## 4 CONCLUSION

This work provides the first evidence that disentanglement-focused representation learning and the resulting models are powerful measurement tools with immediately applicability to applied and basic research for clinical psychology and electrophysiology. One caveat, however, should be considered: more work is needed to further explore how this approach could be used to produce truly meaningful clinical inferences. Toward that goal, this study is limited because of (1) the choice to focus analysis on the superordinate depression class as opposed its constituent disorders, (2) the use of Axis 1 as a point of comparison as opposed to a similarly represented co-morbid disorder, and (3) the choice to train models without between-participant cross-validation. Clearly, there is a need for follow-up work to examine factors that have more clinical impact such as differences in types of depression (persistent vs. major depressive disorder) and treatment response.

We have demonstrated that disentangled representation learning can be successfully applied to

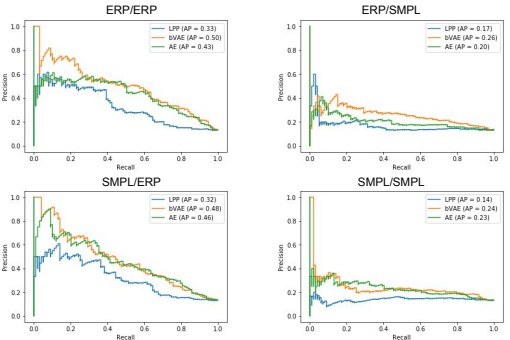

Figure 5: Precision-recall curves for balanced classification accuracy of depression diagnosis calculated as the average of precision and recall for L2 regularised logistic regression. ERP - averaged trajectories, SMPL - single sampled EEG trajectories. LPP - canonical late positive potential baseline representation. Models trained on single EEG trajectories (SMPL) are tested on different single EEG trajectories in the SMPL/SMPL train/test case.

EEG data, resulting in representations that can
be successfully re-used to predict multiple clinical factors through fast supervised re-mapping, out-performing a baseline typical for the field. Our method recovers a representation that can be used to automatically extract denoised Event Related Potentials (ERPs) from novel, single EEG trajectories. Finally, single factors of the learned disentangled representations often correspond to meaningful markers of clinical factors (as automatically detected by SCAN), allowing for human interpretability and potentially providing novel insight into new biomarkers and the neurofunctional alterations underlying mental disorders. While SCAN does not always produce statistically-reliable accuracy advantages over more traditional methods (all using $\beta$-VAE encoding), the ability to show the user exactly how patterns in the raw data manifest across clinical groups of interest is a considerable advantage.

### ACKNOWLEDGMENTS

We thank Sarah Laszlo, Gabriella Levine, Phil Watson, Obi Felten, Mustafa Ispir, Edward De Brouwer, the Amber team, Nader Amir and the Center for Understanding and Treating Anxiety, Dr. Kristen Schmidt, Alec Bruchnak, and Nicholas Santopetro.

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
