# OpenReview forum: "Representation learning for improved interpretability and classification accuracy of clinical factors from EEG"
_ICLR.cc/2021/Conference — ICLR 2021 Poster_

### Official Review · AnonReviewer3 · 2020-10-26
**Solid application of well-selected method to difficult application domain with interesting results - Recommend acceptance**

**Rating:** 7
**Confidence:** 4

**Review:**


POST REVISION

Following our discussions and taking the changes made to the manuscript into account, I have decided to increase my score and recommend acceptance. My concerns have been adequately addressed. I believe the paper has been clarified, results are more carefully evaluated and claims are sound. I believe that this work consitutes a well-selected application that addresses a relevant research question with important clinical implications. In my opinion, this deserves to be aknowledged and may be of interest to others in the ICLR audience. My hope is that this application stimulates more work in EEG-based machine learning and also encourages others not to shy away from difficult application domains such as psychiatry, where we really are in need of new solutions to old and - unfortunatley - quite persistent problems.

Thank you very much for your hard work!







---------------------------------------------------------------------------
The authors propose a disentanglement approach (bVAE with/without SCAN) to obtain sparse and interpretable clinical features from EEG time series to aid clinical decisions and automatize EEG preprocessing. They obtain convincing disentangled representations that show some promise for future applications.

 In the following, I point out strengths and weaknesses, my recommendation and its justification as well as additional detailed comments for the authors. I hope you will find my comments helpful and constructive.

---------------------------------------------------------------------------
Strengths and weaknesses

Strengths
-	Providing ML solutions for psychiatry is a very challenging problem, as psychiatric diseases are of a very complex nature and many crucial processes remain opaque at best. I believe that it is very important that the ML community starts to tackle these issues. I thank the authors for trying to solve this difficult real-world problem and not showcasing their approach on toy examples. Even though the overall accuracies are not high, I believe it is very important to show such an unbiased estimate of where we are at with ML applications in psychiatry. This merits recognition.
-	Choosing EEG data for applications in psychiatry is a very good choice given its feasibility, cost-efficiency and availability in a clinical setting. Often authors do not think carefully enough, whether the data they train their classifier on can actually be acquired in a real-world clinical setting.
-	I applaud the authors’ choice to emphasize interpretability in this healthcare setting. This is a commendable choice that is often not made in other healthcare applications. This is a crucial prerequisite to improve the chance of such an approach ending up in clinical practice as interpretability will be the precursor for acceptance by patients and medical staff.
-	Including study site prediction was a very commendable effort to assess geographic confounders, which is a major concern in recent psychiatric studies with the advent of more and more large scale multi-center datasets.

Weaknesses:
-	Depression outcome label appears to have a proof-of-concept character, rather than addressing a more relevant clinical question (differential diagnosis for bipolar vs unipolar depression for example).
-	Misleading representation of results speaking to superiority of their approach (SCAN approach is much worse than LR combined with bVAE and VAE; therefore, I would advise a more cautious interpretation with respect to the SCAN results and being more precise in the abstract).
-	Confidence intervals for classification results are missing.
-	The claim that their approach reduces ‘hand-engineering’ seems to be unsubstantiated given the description of their methodology.

---------------------------------------------------------------------------
Recommendation
-	Overall, I would like to see this paper accepted, should the authors agree to address the concerns raised above. Unfortunately, I cannot support acceptance as it stands, but I would increase my score, if my concerns have been addressed.

---------------------------------------------------------------------------
Justification of recommendation
-	I believe the research question addresses a very important problem in trying to identify sparse and interpretable clinical markers from EEG data. The method is well-selected to address this question and takes real-world clinical constraints into account (multi-center data, feasibility of data acquisition => EEG, interpretability of identified features => bVAE, uncertainty regarding clinical labels and about which features are required => transfer learning to new labels and unsupervised feature identification). Therefore, I believe this paper deserves to be accepted and represents a good example that hopefully will inspire others to tackle these challenging questions.
-	However, I cannot support acceptance as it stands, because of the weaknesses highlighted above. Most of the issues pertain to clarity of writing, over-interpretation (or at least misleading representation of the results) some missing information on methodology, and choice of the clinical outcome label. For more detailed elaboration on these points see below.

---------------------------------------------------------------------------
Detailed feedback

Abstract
-	You describe your method (bVAE and SCAN) followed by the sentence: “We demonstrate that our method is able to outperform the canonical hand-engineered baseline classification method on a number of factors, including participant age and depression diagnosis”. This gives the impression that the bVAE+SCAN approach is substantially superior to the hand-engineered baseline. However, in the results table this specific combination is only marginally better than baseline, while bVAE+LR or indeed AE+LR substantially outperform the baseline. This is misleading in the abstract and should also be discussed later on.


Introduction
-	You make point to criticize the ‘hand-engineering’ of conventional EEG analysis (section 1, paragraph 2, p. 1-2). You fail to mention that automatic options for eye blink correction exist, for example PCA based methods like Berg & Scherg (1994): https://doi.org/10.1016/0013-4694(94)90094-9 Indeed fully automated pipelines do exist.
Furthermore, from your methods description (2.1, EEG preprocessing, p. 4;  2.1 Autoencoder, last paragraph p. 4-5), it appears that you follow the conventional preprocessing and only then feed the EEG into the autoencoder, which is of course, in principle, fine, but I fail to see why you bring up the whole critique of the hand-engineered pipeline. Your approach does not seem to address this issue, or did I miss something?

Methods
-	Some aspects of the methodological description of the analysis are missing. Please, state which software was used to perform the preprocessing. Were ICA-based eye blink correction results visually inspected? Furthermore, for the AE you state that preprocessed trialwise EEG data goes into it as input, for the bVAE this information is missing. Did you follow this approach here as well?
-	The choice of your clinical classification label appears arbitrary and in my opinion does not address a clinically important question. Could you elaborate why you lumped these specific diagnoses together and not others? The axis I label is so broad that it is basically meaningless. What was the rationale for this analysis? I also assume that you classify depression and axis I vs. controls, is this correct? If so, this is not very relevant clinically. The challenge does not lie in determining, whether a person is healthy or has a mental disorder (clinicians usually can tell within a few minutes of conversation), but rather differential diagnosis or prognosis. A more clinically relevant comparison would be, for example, classifying MDD vs bipolar depression. This is a challenging clinical question, because both patient groups can present with depressive symptoms initially and only time tells which diagnose and also (importantly) medication is appropriate. Why did you not choose such a classification problem?
-	In your description of the bVAE you state that each disentangled factor corresponds to an ‘interpretable’ transformation of the data. This is misleading. Interpretability is of course the goal of a reduced latent space, but by no means the default result. Whether a compression is indeed interpretable needs to be carefully assessed.
-	Was test data selected across subjects (other individuals to test) or within subjects (other parts of the data from the same individual, but all individuals included in the training data)? This is important information as the goal would be to generalize to unseen individuals rather than unseen data from the same individual.

Results
-	Please, add confidence intervals for balanced accuracy (for example by computing the posterior balanced accuracy: Broderson et al. 2010, https://ieeexplore.ieee.org/document/5597285/) otherwise it is impossible to assess the relative quality of the different classifiers. Statements such as: “the classification accuracy is still significantly higher for deep representations compared to the LPP” should be backed up by a statistical test of some sort. Along the same lines how was “significantly higher than chance“ assessed? By permutation tests, based on confidence intervals/Bayesian confidence intervals?
-	You state: “This suggests that replacing the more manual canonical LPP pipeline with deep representation learning can allow for both better training data efficiency and a reduction in time that the (potentially vulnerable) participants have to spend in the lab by up to 37x,…” For this interpretation to hold, you would need to show, that you can train your classifier on a single trial, otherwise, you would still need all the data for pretraining to then be able to make predictions based on single trials. I would advice clarifying this.
-	Lastly, I would like to state, that I was very impressed that you were able to re-discover the anhedonia neurocorrelates with your approach. I think the major challenge is indeed to make sure, that your bVAE identifies meaningful latent variables which I believe you successfully showed.

---

> ### Author Response · Authors · 2020-11-17
> **Response**
>
> Dear Reviewer,
>
> Thank you for your kind words and thoughtful feedback. We have addressed your comments:
>
> - We have updated the manuscript to clarify our point on hand-crafted pre-processing pipelines.
>
> - Which software was used to perform the preprocessing? An automated pipeline was built with MNE (reference added to manuscript).
>
> - “Were ICA-based eye blink correction results visually inspected?”: ICA-based eye blink corrections were not manually inspected. Instead we always removed the first ICA component.
>
> - “For the AE you state that preprocessed trialwise EEG data goes into it as input, for the bVAE this information is missing. Did you follow this approach here as well?” - Yes, we have updated the manuscript to indicate this.
>
> - "The choice of your clinical classification label appears arbitrary and in my opinion does not address a clinically important question." We agree the field is in need of greater clarity with respect to pharmaceutical treatment outcomes or MDD vs. PDD as another example. The main impediment to an approach like this is that it is much more difficult to recruit these rarer participant splits at ML scale. The logic behind focusing on MDD and providing the broad umbrella label of Axis 1 was to explore a transdiagnostic "all takers" approach and maximize N for training and evaluation. In our view, this tradeoff between clinical relevance and generalizable representation learning was appropriate considering that the goal was to test that the disentanglement pipeline could preserve any relevant signal in a sparsely-labeled dataset with a purely unsupervised learning regime.
>
> - The reason why we claim interpretability for the beta-VAE latents is because we know that well disentangled models are almost always interpretable when applied to image data. This is indeed the goal of the disentangling objective. Rather than pure compression, which is the goal of AE, beta-VAE attempts to find semantically meaningful latents. While disentangled latents obtained from EEG data may not immediately look interpretable, we have verified that they are indeed such, because SCAN was able to attach interpretable labels to single latents of pre-trained beta-VAE models.
>
> - While we agree that ideally we would like to generalise across objects, the current results are presented in the within object scenario. We have launched new experiments to check whether the same pattern of results would hold across objects and will update you if we manage to produce results by the time the discussion period is over.
>
> - Thank you for pointing out the Broderson et al. 2010 reference to us. We have updated our results with the bayesian estimate on the balanced classification accuracy, which indicates that our results are statistically significant in the majority of cases (both against chance and against the LPP baseline) as shown in Figure 2.
>
> - We actually do train our models on single trials (the SMPL/* condition in Figure 2). Furthermore, since our approach is currently addressing within object generalisation, we envisage that a clinician would record more data from each object during the first visit to obtain ERPs and add those to the training data for the models, but subsequently these objects could be diagnosed from single trajectories.

---

> > ### Comment · AnonReviewer3 · 2020-11-19
> > **Response to changes made to the manuscript**
> >
> > Thank you very much for your hard work. Here are some remaining comments.
> >
> > - We have updated the manuscript to clarify our point on hand-crafted pre-processing pipelines.
> > I fail to see, where you made these changes. In the latest version I downloaded just now, I do not see any changes with regard to your claims about hand-crafted pre-processing in the introduction. You still fail to aknowledge that automatized options exists and ARE already in use, like the Berg and Scherg eyeblink correction method or automatic thresholds as implemented in the SPM software package (see for example: Weber et al. DOI: https://doi.org/10.1523/JNEUROSCI.3069-19.2020)
> > I believe that you are critcizing something that is already being addressed or indeed solved with regard to the hand-crafting and would recommend to reign in your claims here. (This does not mean that you are not contributing with the bVAE and SCAN of course).
> >
> > - Choice of clinical labels
> > Thank you very much for explaining your choice of clinical classification label. Given the constraints of the data that may be the best you can do, however, this should enter limitations with a note to encourage others to move towards more relevant clinical classification problems. Furthermore, given that already two of us were having trouble with understanding the rational with respect to the axis I prediction you should maybe clarify this in the manuscript.
> >
> > - The reason why we claim interpretability for the beta-VAE latents is because we know that well disentangled models are almost always interpretable when applied to image data.
> > First of all almost always is not always and secondly EEG data may be different from other imaging data (it is inherently less intuitive and intepretable). I would advise phrasing this more carefully.
> >
> > - We have updated our results with the bayesian estimate on the balanced classification accuracy, which indicates that our results are statistically significant in the majority of cases (both against chance and against the LPP baseline) as shown in Figure 2.
> > Thank you for incorporating this change. The fact that SCAN does not seem to add much more in terms of balanced accuracy compared to a simple LR+bVAE could be discussed in the paper along the lines of your response to RV 5.
> >
> > - While we agree that ideally we would like to generalise across objects, the current results are presented in the within object scenario. We have launched new experiments to check whether the same pattern of results would hold across objects and will update you if we manage to produce results by the time the discussion period is over.
> > Your cross-validation strategy and the consequences you mentioned in your response to me should be clarified in the text, so this point becomes clear to the reader, possible also adding the limitations this decision entails.
> >
> >
> > PS: Referring to patients as ‘objects’ should be avoided (e.g. in 'within object generalisation'), rather refer to them as patients or individuals. I do say this to annoy you, but it is important to remember that these models, however good or bad they are have real-world consequences for people (not objects), which is especially important in a clinical setting, where ethical concerns are always a consideration.

---

> > > ### Author Response · Authors · 2020-11-21
> > > **Updated prose on interpretability, preprocessing, clinical relevance of labels, and use of ICA and other preprocessing. Inclusion of ICA-free classification differences.**
> > >
> > > - Apologies for the error---you're correct that the revised manuscript did not include our intended changes. Again, thanks for raising this issue and allowing us to clarify.
> > >
> > >      In noting the limitations of 'hand engineered' features, we did not intend to solely refer to data pre-processing and artifact rejection/correction but to the entire pipeline exemplified by the typical approach of relying on a priori definitions of ERP components to encode the putative biomarker. For example, in a conventional study, the LPP response would be quantified as the average amplitude within a pre-specified time window and over specific electrodes. It is primarily this conventional approach that we have in mind when contrasting bVAE and SCAN with the traditional 'hand-engineered approach'.
> > >
> > >      As for the issue of automating the post-processing handling of EEG, we did not intend to dispute that there are other fully automated procedures in common usage and we do not mean to imply that ours is the first of this kind. We have now clarified this in the manuscript and make it clear that our approach provides an incremental contribution to this field rather than a de novo invention. Lastly, we include here an early snapshot of our results without ICA (Table A6 in the updated manuscript) that show that the outcome is resilient against the presence of uncorrected artifacts. (We're currently revising the manuscript to present ICA-free data and results.)
> > >
> > > - Axis 1: We have amended the manuscript to identify this potential limitation of the study and further explained the intention behind including Axis 1 as a point of comparison.
> > >
> > > - We've taken the interpetability issue raised here under consideration. We have no reason to suspect disentanglement is working less well in this domain. It is our view that it would not be possible to recover or interpret the LPP without deep a priori knowledge of ERP componentry and its phenomenology in clinical populations, i.e., the exact issue we hope the disentanglement pipeline can help address. With the use of SCAN, encoded EEG can be mapped to labels of interest and the resulting reconstructions can be compared visually and with statistical inference to (re)discover clinically important differences. Saying this, we have softened the claims of interpretability throughout the paper by adding modifiers, like “often” and “arguably”.
> > >
> > > - We take your point about SCAN and it's accuracy advantage relative to the other conditions. That said, none of the other conditions allow for the projection of discrete labels back through the raw data encoder for the purpose of explainability (and potentially novel biomarker discovery). We suggest these advantages make a clear case for SCAN where accuracy performance is better or not reliably different from the compared conditions and the result is a tool that can show you the properties that separate different groups in latent space and how those properties manifest in the raw data. We have added a sentence in the conclusion to address this point.
> > >
> > > - The manuscript has been amended to clarify the cross-validation approach. We totally agree about the language used to describe participants, thanks for raising the concern.

---

> > > > ### Comment · AnonReviewer3 · 2020-11-24
> > > > **All concerns addressed**
> > > >
> > > > Thank you very much! All of my concerns have been addressed. I am willing to raise my score and recommend acceptance now. Thank you for your hard work and for investing your time in this interesting and relevant application.

---

### Official Review · AnonReviewer1 · 2020-10-29
**Review #1**

**Rating:** 6
**Confidence:** 4

**Review:**

Summary: The authors propose a beta-VAE network to learn EEG representation as biomarkers for diagnosing depression from EEG data. They show improved performance compared to an off-the shelf linear classifier. The paper is well-written but lacks a description of related work in the field and also a detailed analysis of the results to support the claims.

Novelty:  The use of VAE and beta-VAE for EEG data is not novel and this line of literature should be better discussed in the paper.

A few more comments/questions for the authors:

1. The details of the AE and beta-VAE architecture should be described in the main paper not the supplementary. Also, a detailed description of the number of trainable parameters and the amount of available data should be added to the main manuscript.

2. On page 2, the authors mention that LDA and SVM are not commonly used in EEG literature for ERP classification, but these two are actually very common. For instance see: Blankertz, B., Lemm, S., Treder, M., Haufe, S., & Müller, K. R. (2011). Single-trial analysis and classification of ERP components—a tutorial. NeuroImage, 56(2), 814-825.

3. The authors mention that the EEG “ground truth” markers are not available for a condition like depression, and yet they mention that their method provides interpretable biomarkers. There seems to be some discrepancy here that should be explained further.

4. The authors mention that their paper is the first to use DL representation learning for clinical EEG biomarkers. This claim is not true, please see the following for a list of relevant papers: Roy, Y., Banville, H., Albuquerque, I., Gramfort, A., Falk, T. H., & Faubert, J. (2019). Deep learning-based electroencephalography analysis: a systematic review. Journal of neural engineering, 16(5), 051001.

5. The axis 1 label is rather confusing, Could the authors please explain it further and say how it is relevant for the depression diagnosis study?

6. Did the authors do pre-processing (including ICA and removing eye-blinks) at test time? ICA needs a lot of EEG data to be able to reliably remove the eye blinks. That means a pre-trained beta-VAE would not be able to reduce training time.

7. The authors do not mention which electrodes are selected for their analysis and why.

8. It is not clear if LR is even significantly better than the other baseline classifier in table A2. Also, the authors should apply the rest of the baseline classifiers to the beta-VAE and AE representations.

9. In table 1 and A2, the upper limit of the chance level should be calculated as a function of the available data as described in: Müller-Putz, G., Scherer, R., Brunner, C., Leeb, R., & Pfurtscheller, G. (2008). Better than random: a closer look on BCI results. International Journal of Bioelectromagnetism, 10(ARTICLE), 52-55.
In the same table, the overall chance is reported as 0.45. Why is that?

10. How did the authors evaluate the “significantly better” performance of their proposed classifier?

11. On page 7, how is the 37x less time need to be spent in the lab by vulnerable populations measured? Again, did the authors used ICA-cleaned data for this evaluation?

12. Is the separation of the ERPs in Figure 3C significant? Please provide the original ERP for the depression and healthy subjects for comparison.
Also, the two colors are very hard to tell apart, please use another color combination such as blue and red/orange.

---

> ### Author Response · Authors · 2020-11-17
> **Response**
>
> Dear Reviewer,
>
> Thank you for your thoughtful feedback. We have addressed your comments:
>
> - We have moved AE and beta-VAE architecture descriptions to the main paper and included the count of parameter numbers. We have also added the amount of trainable data to the appendix.
>
> - We have removed the claim that LDA and SVM are not common in the field.
>
> - “EEG “ground truth” markers are not available for a condition like depression, and yet they mention that their method provides interpretable biomarkers”: “ground truth” in our paper refers to the classification labels. We claim that these are not available, because there is typically a lot of variance in the diagnostic labels between different clinicians. Hence, the classification labels that are available are not “ground truth” but instead are a noisy approximation of the “ground truth”. One cannot unambiguously say that a certain EEG signal should be labelled as “depressed” in the same way as one can unambiguously say that a picture of a cat should be labelled as “cat”. On the other hand, it is widely believed that EEG data should contain biomarker signals for clinical disorders, but as a field we do not know what all of them are yet. Our method attempts to discover such biomarkers.
>
> - Thank you for pointing us towards the Roy et al (2019) work. We have cited it in our paper and removed the claim about deep representation learning.
>
> - Regarding the use of the Axis 1 diagnostic label: We included this transdiagnostic indicator of psychopathology instead of using the relatively small n of another disorder for comparison. Axis 1 includes the most prevalent psychological disorders in the population. Further, given high levels of comorbidity, this was a way to model the presence of any psychiatric illness. It is provided to give the reader a sense of what the algorithm may have learned that is generalizable across disorders---essentially something like the P factor (Caspi, Houts, Belsky, et al., 2014). The data collection for the study was primarily focused on depression but given that the SCID produces a large set of diagnostic decisions, we collapsed the sparser set of positively-diagnosed individuals into the existing DSM-IV superordinate category Axis 1.
>
> - In the current implementation ICA pre-processing to remove eye blinks was indeed applied at test time. This would still help with better data efficiency, since per-participant ICA components can be stored and re-used for subsequent data analysis. Hence, while the initial recording session for each participant may be of the same length as is typical in the field, subsequent visits could be limited in time. Saying this, we are currently re-running our pipeline using raw EEG data with no ICA pre-processing and will report these results as soon as they are available.
>
> - Regarding electrode choice: The analysis used EEG collected at Fz, Cz, Pz midline electrode sites. Only three electrodes were used because the project was de-risking the simplest possible cap setup so that the technology would be as easy to deploy as a cardiac stress test in the office of a general practitioner. The LPP is traditionally indexed from centro-parietal midline electrodes so this minimal electrode setup was used to maximize the speed of application and ease of use.
>
> - We have tempered the claim that LR is better than the other classifiers in the LPP case, and included the results from all other classifiers applied to beta-VAE and AE representations in the Appendix (Tbls A3-A4 and Figs A3-A4)
>
> - We have updated our classification results with the bayesian estimate on the posterior distribution of balanced classification accuracy as per Broderson et al. 2010, https://ieeexplore.ieee.org/document/5597285/, as suggested by Reviewer 3 (see Fig 2 in the updated manuscript). These results now also include 95% confidence intervals, which demonstrate that most of the results are significantly above chance. Also beta-VAE and AE significantly outperform the LPP baseline in most cases.
>
> - We have updated Figure 3C (now Figure 4C) to add original ERP samples for comparison with the SCAN samples. We have also updated all figures to replace the green colour with red as per your suggestion.
>
>
> Caspi, A., Houts, R. M., Belsky, D. W., Goldman-Mellor, S. J., Harrington, H., Israel, S., ... & Moffitt, T. E. (2014). The p factor: one general psychopathology factor in the structure of psychiatric disorders?. Clinical Psychological Science, 2(2), 119-137.

---

> > ### Author Response · Authors · 2020-11-21
> > **Added results without ICA pre-processing**
> >
> > Dear Reviewer,
> >
> > We would like to let you know that we have obtained preliminary classification results using our pipeline on data that does not undergo any ICA-based pre-processing, as shown in Table A6 in the updated manuscript. On average it appears that removing this pre-processing slightly hurts the traditional LPP baseline, while on the contrary slightly improving the results of our proposed pipeline, with SCAN+bVAE seeing the most improvement.

---

> > > ### Comment · AnonReviewer1 · 2020-11-23
> > > **Response to the recent changes**
> > >
> > > Thanks a lot for your response and the updated results.
> > >
> > > I have another question regarding the number of trainable parameters. This is huge! And it is not clear if you really have enough data to train reasonable EEG patterns instead of simply noise. I believe the fact that your method improves without ICA pre-processing may also be due to the overfitting issue. Could you please elaborate on that?
> > >
> > > Also, what statistical test have you used?
> > > In figure 2, the upper limit of the chance level should be calculated as a function of the available data as described in: Müller-Putz, G., Scherer, R., Brunner, C., Leeb, R., & Pfurtscheller, G. (2008). Better than random: a closer look on BCI results. International Journal of Bioelectromagnetism, 10(ARTICLE), 52-55.

---

> > > > ### Author Response · Authors · 2020-11-24
> > > > **Added confidence intervals to chance and responded to the overfitting concern**
> > > >
> > > > Dear Reviewer,
> > > >
> > > > Thank you for raising the overfitting question. While it is a valid concern, we believe that our pipeline is not overfitting for the following reasons:
> > > >
> > > > 1) All of the results presented in the paper, apart from the ERP/ERP scenario, use different EEG samples for training and testing.
> > > >
> > > > 2) Neither beta-VAE/AE nor SCAN are trained for classification. Instead they are trained for reconstruction of their corresponding data. When we do repurpose the pipeline to perform classification, we do so by passing the data through a pre-trained beta-VAE/AE encoder and then through the pre-trained SCAN decoder - a combination that was never trained together (beta-VAE/AE is trained separately from SCAN, then fixed during SCAN training, and SCAN decoder is only optimised with respect to the SCAN (not beta-VAE/AE) encoder). Hence, it is not obvious to us how these models can overfit to the classification objective.
> > > >
> > > > 3) Some of our best classification accuracy results are obtained with the beta-VAE/AE + linear regression combination. In this case, none of the representation learning models (beta-VAE or AE) were exposed to any training labels at all, and hence could not possibly overfit to the classification task.
> > > >
> > > > 4) Since our models are optimising for reconstruction, and because each reconstructed “pixel”/sample is parameterized by an independent Bernoulli distribution, our learning objective can effectively be decomposed into 256x6=1,536 independent learning objectives per EEG “image” and 11 independent learning objectives per SCAN label. This means that the effective data size then becomes 1536x758=1,164,288 for beta-VAE/AE training and 11x758=8,338 for SCAN. This means that beta-VAE/AE definitely cannot overfit to the data given the model’s approximately 106,752 parameters. SCAN does have more parameters (approximately 22,016) than its effective data size, however this is very common in modern deep learning (e.g. see Zhang et al, 2017 https://arxiv.org/pdf/1611.03530.pdf: "The number of parameters exceeds the number of data points as it usually does in practice"). As is discussed in the same reference, it appears that such over-parametrisation does not typically result in overfitting in deep learning ("Despite their massive size, successful deep artificial neural networks can exhibit a remarkably small difference between training and test performance"). While the precise reason for the lack of overfitting in deep learning remains a mystery, the authors of the cited paper attribute the surprising generalisation of deep neural networks to the implicit regularization of stochastic gradient descent.
> > > >
> > > > In terms of the bounds on the chance classification accuracy, we have now included these in Figure 2 (thank you for pointing the reference to us again). As you can see, the majority of our results are indeed significant---the representations obtained from beta-VAE (or AE unless used in combination with SCAN) tend to produce higher accuracy than chance. This holds for all conditions for Age and Site prediction, and for ERP-trained models used in combination with LR in Gender, Depression and Axis 1 conditions. Sample trained beta-VAE models (used with LR or SCAN) are also above chance level for classifying depression. We have updated the text to modify our claims where appropriate.

---

### Official Review · AnonReviewer5 · 2020-11-04
**An extensive and rigorous validation of the power of autoencoders for EEG based classification, but with little novelty**

**Rating:** 7
**Confidence:** 3

**Review:**

Summary

The authors are concerned with the classification of EEG signals, in order to predict age, gender, depression and Axis 1 disorder diagonosis from EEG signals. After standard preprocessing and optionnal averaging to obtain evoked responses, the authors feed the samples into a $\beta$-VAE , and then either use a standard classification algorithm or the SCAN method to predict the labels.
The authors report better results than the usual methods based on the late positive potential. They also show that their method can be trained with non-averaged EEG data and still yield good results when tested on ERP , and conversely. Finally, the authors inspect the learned representations.

Major comments

- The paper is very well written, easy and pleasant to read, and well structured.
- The automation of EEG pipelines, like this paper does, is extremely important.
- The SCAN + $\beta$-VAE or SCAN+ VAE method does not seem to perform much better than LR +LPP. Even though SCAN allows for interpretable components, it is arguably much less interpretable than LPP.
- The article validates carefully  a machine learning pipeline on a specific task and dataset, but there is little contribution in terms of machine learning, so I'm wondering whether ICLR is a good fit for this paper, rather than a more neuroscience-oriented conference.

Minor comments
- The authors propose an original pipeline, yet the dataset and the code to reproduce the results are not provided, which hinders reproducibility and the potential impact of this work.
- In my understanding, the LPP seems to only use one feature for classification: the average amplitude difference between waveforms. Could the authors also consider methods using more hand-crafted features?
- The EEG signals are only acquired with 3 sensors, it would be interesting to add a word about how the method scales to datasets with more sensors.
-  It would be interesting to add some ROC curves for the logistic regressions, which would complement nicely the summary statistic used by the authors.

Misc.

- The software used to perform the study should be acknowledged.
- The ERP are normalized to [0, 1] before going in the VAE. The authors could be more accurate: is each channel normalized individually?
- The references that state that EEG contains important biomarkers of clinical disorders, and that averaging trials yields ERP could also point to more historic papers.

---

> ### Author Response · Authors · 2020-11-17
> **Response**
>
> Dear Reviewer,
>
> Thank you for your thoughtful feedback. We have addressed your comments:
>
> - “The SCAN + β-VAE or SCAN+AE method does not seem to perform much better than LR +LPP”:  you are right that SCAN+AE does not work above chance level -- this is because SCAN was developed to only work with disentangled representations, and AE representations are instead entangled. Saying this, SCAN+beta-VAE does outperform the LR+LPP method (0.55 vs 0.52 average over all classification problems and all conditions). While this difference might not appear large, it is actually statistically significant as demonstrated by the new Bayesian calculations we have introduced as per the suggestions by Reviewers 2 and 3 (see Fig. 2 in the updated manuscript).
>
> - “Even though SCAN allows for interpretable components, it is arguably much less interpretable than LPP”: while it is true that LPP is interpretable, its interpretability comes from the fact that it is engineered by clinicians by utilising the existing knowledge about LPP being a biomarker of depression. In contrast, SCAN provides interpretability through a largely unsupervised deep learning pipeline. This makes SCAN unique and different, since it does not rely on existing knowledge of what biomarkers look like in the EEG signal (as LPP does), and instead it only relies on the availability of the current iteration of diagnostic labels. By pairing the provided labels with the representations learnt by beta-VAE, our SCAN+beta-VAE system allows for the discovery of new biomarkers, i.e. the acquisition of new knowledge. We have validated this pipeline by “re-discovering” the LPP biomarker.
>
> - “The article validates carefully a machine learning pipeline on a specific task and dataset, but there is little contribution in terms of machine learning”: we did consider submitting this work to a neuroscience journal, however we thought that it fit this venue well, targeting the "applications in audio, speech, robotics, neuroscience, computational biology, or any other field" ICLR call for papers. While we acknowledge that the methodology in itself presented in our work is not new, its adaptation to interpretable classification of clinical factors from EEG signal is not obvious and is novel.
>
> - “The dataset and the code to reproduce the results are not provided”: if the paper is accepted, we will open-source part of the dataset and the code for implementing the models.
>
> - “Could the authors also consider methods using more hand-crafted features?”: We agree that the LPP is only one of a larger set of hypothesized biomarkers that have potential for measuring and diagnosing depression and that this larger feature set would likely be more effective---for e.g., information theoretic approaches across the entire time series (perhaps even outside of the ERP framework with raw EEG or resting-state EEG measurements). However, it would also change the nature of the work from an inquiry directed at re-discovering an existing biomarker to optimizing for more accurate diagnosis. This would be a hard pivot away from the key goal of this work: take an established empirical method for eliciting a known, stereotyped neural response and rediscover the "condition delta" effect between neutral and emotionally evocative stimuli with a purely unsupervised approach specifically focused on explainability.
>
> - “It would be interesting to add a word about how the method scales to datasets with more sensors”: the nice thing about our pipeline is that it is very general. This means that it will easily scale to more sensors, simply by extending the ‘images’ used as input to the beta-VAE/AE from 6xT to NxT, where N is the new number of channels and T is the number of time steps.
>
> - “It would be interesting to add some ROC curves for the logistic regressions”: we have added precision-recall curves (since our data is unbalanced) to the appendix. Please see Figures 3 and A2-A4 in the new manuscript.
>
> - Misc: we have updated the appendix to acknowledge the software used to perform the study. The ERPs were normalised to [0, 1] across all channels together, not individually.
>
> - What historic papers would you recommend that we cite with regards to biomarkers of clinical disorders in EEG?

---

> > ### Comment · AnonReviewer3 · 2020-11-19
> > **Little Contribution to Machine Learning - Comment**
> >
> > In my opinion, I do believe that a well-selected and rigorous application of state-of-the-art methods to a problem that is relevant to society (here in the field of mental health) is indeed an important contribution and should not be disregarded. Good applications may serve as inspiration for other scientists and health professionals and help to bring the progress that is being made in machine learning into the clinical setting, which is very important in my opinion. I would therefore caution against dismissing applications too easily.

---

### Decision · Program_Chairs · 2021-01-07
**Final Decision**

**Decision:**

Accept (Poster)

**Comment:**

The approach is novel and according to the reviewers' comments addresses a relevant and important problem on EEG data analysis. Differences to related work are discussed. Methods and Experimental results are sound. The authors have provided a comprehensive response to the reviews.